# Loaded DiCE: Trading off Bias and Variance in Any-Order Score Function Estimators for Reinforcement Learning

**Gregory Farquhar** *
University of Oxford

**Shimon Whiteson**
University of Oxford

**Jakob Foerster**
Facebook AI Research

## Abstract

Gradient-based methods for optimisation of objectives in stochastic settings with unknown or intractable dynamics require estimators of derivatives. We derive an objective that, under automatic differentiation, produces low-variance unbiased estimators of derivatives at any order. Our objective is compatible with arbitrary advantage estimators, which allows the control of the bias and variance of any-order derivatives when using function approximation. Furthermore, we propose a method to trade off bias and variance of higher order derivatives by discounting the impact of more distant causal dependencies. We demonstrate the correctness and utility of our objective in analytically tractable MDPs and in meta-reinforcement-learning for continuous control.

## 1 Introduction

In stochastic settings, such as reinforcement learning (RL), it is often impossible to compute the derivative of our objectives, because they depend on an unknown or intractable distribution (such as the transition function of an RL environment). In these cases, gradient-based optimisation is only possible through the use of stochastic gradient estimators. Great successes in these domains have been found by building estimators of first-order derivatives which are amenable to automatic differentiation, and using them to optimise the parameters of deep neural networks [François-Lavet et al., 2018].

Nonetheless, for a number of exciting applications, first-order derivatives are insufficient. Meta-learning and multi-agent learning often involve differentiating through the learning step of a gradient-based learner [Finn et al., 2017, Stadie et al., 2018, Zintgraf et al., 2019, Foerster et al., 2018a]. Higher-order optimisation methods can also improve sample efficiency [Furmston et al., 2016]. However, estimating these higher order derivatives correctly, with low variance, and easily in the context of automatic differentiation, has proven challenging.

Foerster et al. [2018b] propose tools for constructing estimators for any-order derivatives that are easy to use because they avoid the cumbersome manipulations otherwise required to account for the dependency of the gradient estimates on the distributions they are sampled from. However, their formulation relies on pure Monte-Carlo estimates of the objective, introducing unacceptable variance in estimates of first- and higher-order derivatives and limiting the uptake of methods relying on these derivatives.

Meanwhile, great strides have been made in the development of estimators for first-order derivatives of stochastic objectives. In reinforcement learning, the use of learned value functions as both critics and baselines has been extensively studied. The trade-off between bias and variance in gradient estimators can be made explicit in mixed objectives that combine Monte-Carlo samples of the

objective with learned value functions [Schulman et al., 2015b]. These techniques create families of *advantage estimators* that can be used to reduce variance and accelerate credit assignment in first-order optimisation, but have not been applied in full generality to higher-order derivatives.

In this work, we derive an objective that can be differentiated any number of times to produce correct estimators of higher-order derivatives in Stochastic Computation Graphs (SCGs) that have a Markov property, such as those found in RL and sequence modeling. Unlike prior work, this objective is fully compatible with arbitrary choices of advantage estimators. When using approximate value functions, this allows for explicit trade-offs between bias and variance in any-order derivative estimates to be made using known techniques (or using any future advantage estimation methods designed for first-order derivatives). Furthermore, we propose a method for trading off bias and variance of higher order derivatives by discounting the impact of more distant causal dependencies.

Empirically, we first use small random MDPs that admit analytic solutions to show that our estimator is unbiased and low variance when using a perfect value function, and that bias and variance may be flexibly traded off using two hyperparameters. We further study our objective in more challenging meta-reinforcement-learning problems for simulated continuous control, and show the impact of various parameter choices on training. Demonstration code is available at `https://github.com/oxwhirl/loaded-dice`. Only a handful of additional lines of code are needed to implement our objective in any existing codebase that uses higher-order derivatives for RL.

## 2   Background

### 2.1   Gradient estimators

We are commonly faced with objectives that have the form of an expectation over random variables. In order to calculate the gradient of the expectation with respect to parameters of interest, we must often employ gradient estimators, because the gradient cannot be computed exactly. For example, in reinforcement learning the environment dynamics are unknown and form a part of our objective, the expected returns. The polyonymous "likelihood ratio", "score function", or "REINFORCE" estimator is given by

$$\nabla_\theta \, \mathbb{E}_x[f(x, \theta)] = \mathbb{E}_x[f(x, \theta)\nabla_\theta \log p(x; \theta) + \nabla_\theta f(x, \theta)]. \tag{1}$$

The expectation on the RHS may now be estimated from Monte-Carlo samples drawn from $p(x; \theta)$. Often $f$ is independent of $\theta$ and the second term is dropped. If $f$ depends on $\theta$, but the random variable does not (or may be reparameterised to depend only deterministically on $\theta$) we may instead drop the first term. See Fu [2006] or Mohamed et al. [2019] for a more comprehensive review.

### 2.2   Stochastic Computation Graphs and MDPs

Stochastic computation graphs (SCGs) are directed acyclic graphs in which nodes are determinsitic or stochastic functions, and edges indicate functional dependencies [Schulman et al., 2015a]. The gradient estimators described above may be used to estimate the gradients of the objective (the sum of cost nodes) with respect to parameters $\theta$. Schulman et al. [2015a] propose a *surrogate loss*, a single objective that produces the desired gradient estimates under differentiation.

Weber et al. [2019] apply more advanced first-order gradient estimators to SCGs. They formalise Markov properties for SCGs that allow the most flexible and powerful of these estimators, originally developed in the context of reinforcement learning, to be applied. We describe these estimators in the following subsection, but first define the relevant subset of SCGs. To keep the main body of this paper simple and highlight the most important known use case for our method, we adopt the notation of reinforcement learning rather than the more cumbersome notation of generic SCGs.

The graph in reinforcement learning describes a Markov Decision Process (MDP), and begins with an initial state $s_0$ at time $t = 0$. At each timestep, an action $a_t$ is sampled from a stochastic policy $\pi_\theta$, parameterised by $\theta$, that maps states to actions. This adds a stochastic node $a_t$ to the graph. The state-action pair leads to a reward $r_t$, and a next state $s_{t+1}$, from which the process continues. A simple MDP graph is shown in Figure 1. In the figure, as in many problems, the reward conditions only on the state rather than the state and action. We consider episodic problems that terminate after $T$ steps, although all of our results may be extended to the non-terminating case. The (discounted) rewards are the cost nodes of this graph, leading to the familiar reinforcement learning objective of an

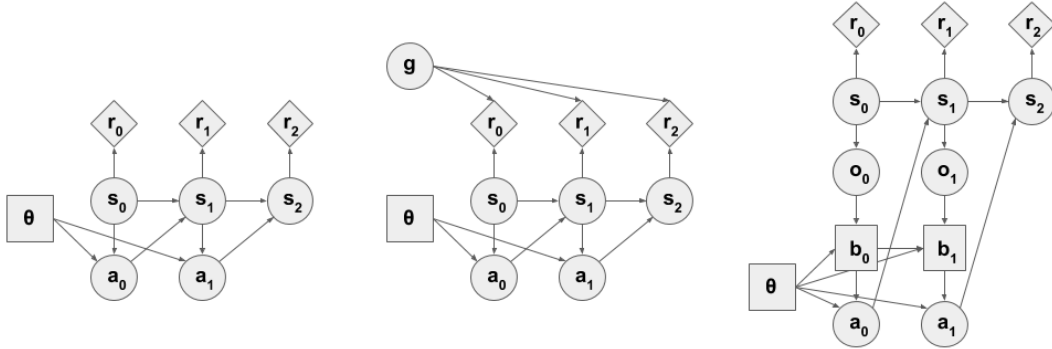

Figure 1: Some example SCGs that support our new objective. From left to right (a) Vanilla MDP (b) MDP with stochastic latent goal variable $g$ (c) POMDP

expected discounted sum of rewards: $J = \mathbb{E}[\sum_{t=0}^{T} \gamma^t r_t]$, where the expectation is taken with respect to the policy as well as the unknown transition dynamics of the underlying MDP.

A generalisation of our results holds for a slightly more general class of SCGs as well, whose objective is still a sum of rewards over time. We may have any number of stochastic and deterministic nodes $\mathcal{X}_t$ corresponding to each timestep $t$. However, these nodes may only influence the future rewards through their influence on the next timestep. More formally, this Markov property states that for any node $w$ such that there exists a directed path from $w$ to any $r_{t'}, t' \geq t$ not blocked by $\mathcal{X}_t$, none of the descendants of $w$ are in $\mathcal{X}_t$ (definition 6 of Weber et al. [2019]). This class of SCGs can capture a broad class of MDP-like models, such as those in Figure 1.

## 2.3 Gradient estimators with advantages

A value function for a set of nodes in an SCG is the expectation of the objective over the other stochastic variables (excluding that set of nodes). These can reduce variance by serving as control variates ("baselines"), or as critics that also condition on the sampled values taken by the corresponding stochastic nodes (i.e. the sampled actions). The difference of the critic and baseline value functions is known as the advantage, which replaces sampled costs in the gradient estimator.

Baseline value functions only affect the variance of gradient estimators [Weaver and Tao, 2001]. However, using learned, imperfect critic value functions results in biased gradient estimators. We may trade off bias and variance by using different mixtures of sampled costs (unbiased, high variance) and learned critic value functions (biased, low variance). This choice of advantage estimator and its hyperparameters can be used to tune the bias and variance of the resulting gradient estimator to suit the problem at hand.

There are many ways to model the advantage function in RL. A popular and simple family of advantage estimators is proposed by Schulman et al. [2015b]:

$$A^{GAE(\gamma,\tau)}(s_t, a_t) = \sum_{t'=t}^{\infty} (\gamma\tau)^{t'-t} \big( r_{t'} + \gamma\hat{V}(s_{t'+1}) - \hat{V}(s_{t'}) \big). \tag{2}$$

The parameter $\tau$ trades off bias and variance: when $\tau = 1$, $A$ is formed only of sampled rewards and is unbiased, but high variance; when $\tau = 0$, $A^{GAE}$ uses only the next sampled reward $r_t$ and relies heavily on the estimated value function $\hat{V}$, reducing variance at the cost of bias.

## 2.4 Higher order estimators

To construct higher order gradient estimators, we may recursively apply the above techniques, treating gradient estimates as objectives in a new SCG. Foerster et al. [2018b] note several shortcomings of the surrogate loss approach of Schulman et al. [2015a] for higher-order derivatives. The surrogate loss cannot itself be differentiated again to produce correct higher-order estimators. Even estimates produced using the surrogate loss cannot be treated as objectives in a new SCG, because the surrogate loss severs dependencies of the sampled costs on the sampling distribution.

To address this, Foerster et al. [2018b] introduce DiCE, a single objective that may be differentiated repeatedly (using automatic differentiation) to produce unbiased estimators of derivatives of any order. The DiCE objective for reinforcement learning is given by

$$J_{\boxdot} = \sum_{t=0}^{T} \gamma^t \boxdot(a_{\leq t}) r_t, \tag{3}$$

where $a_{\leq t}$ indicates the set of stochastic nodes (i.e. actions) occurring at timestep $t$ or earlier.

$\boxdot$ is a special operator that acts on a set of stochastic nodes $\mathcal{W}$. $\boxdot(\cdot)$ always *evaluates* to 1, but has a special behaviour under differentiation:

$$\nabla_\theta \boxdot(\mathcal{W}) = \boxdot(\mathcal{W}) \sum_{w \in \mathcal{W}} \nabla_\theta \log p(w; \theta) \tag{4}$$

This operator in effect automates the likelihood-ratio trick for differentiation of expectations, while maintaining dependencies such that the same trick will be applied when computing higher order derivatives. For notational convenience in our later derivation, we extend the definition of $\boxdot$ slightly by defining its operation on the empty set: $\boxdot(\varnothing) = 1$, so it has a zero derivative.

The original version of DiCE has two critical drawbacks compared to the state-of-the-art methods described above for estimating first-order derivatives of stochastic objectives. First, it has no mechanism for using baselines to reduce the variance of estimators of higher order derivatives. Mao et al. [2019], and Liu et al. [2019] (subsequently but independently) suggest the same partial solution for this problem, but neither provide proof of unbiasedness of their estimator beyond second order. Second, DiCE (and the estimator of Mao et al. [2019] and Liu et al. [2019]) are formulated in a way that requires the use of Monte-Carlo sampled costs. Without a form that permits the use of critic value functions, there is no way to make use of the full range of possible advantage estimators.

In an exact calculation of higher-order derivative estimators, the dependence of a given reward on all previous actions leads to nested sums over previous timesteps. These terms tend to have high variance when estimated from data, and become small in the vicinity of local optima, as noted by Furmston et al. [2016]. Rothfuss et al. [2018] use this observation to propose a simplified version of the DiCE objective dropping these dependencies:

$$J_{LVC} = \sum_{t=0}^{T} \boxdot(a_t) R_t \tag{5}$$

This estimator is biased for higher than first-order derivatives, and Rothfuss et al. [2018] do not derive a correct unbiased estimator for all orders, make use of advantage estimation in this objective, or extend its applicability beyond meta-learning in the style of MAML [Finn et al., 2017].

In the next section, we introduce a new objective which may make use of the critic as well as baseline value functions, and thereby allows the bias and variance of any-order derivatives to be traded off through the choice of an advantage estimator. Furthermore, we introduce a discounting of past dependencies that allows a smooth trade-off of bias and variance due to the high-variance terms identified by Furmston et al. [2016].

## 3 Method

The DiCE objective is cast as a sum over rewards, with the dependencies of the reward node $r_t$ on its stochastic causes captured by $\boxdot(a_{\leq t})$. To use critic value functions, on the other hand, we must use forward-looking sums over returns.

This is possible if the graph maintains the Markov property defined above in Section 2.2 with respect to its objective, so as to permit a sequential decomposition of the cost nodes, i.e., rewards $r_t$, and their stochastic causes influenced by $\theta$, i.e., the actions $a_t$. We begin with the DiCE objective for a discounted sum of rewards given in (3), where our true objective is the expected discounted sum of rewards in trajectories drawn from a policy $\pi_\theta$.

We define, as is typical in RL, the return $R_t = \sum_{t'=t}^{T} \gamma^{t'-t} r_{t'}$. Now we have $r_t = R_t - \gamma R_{t+1}$, so:

$$J_{\odot} = \sum_{t=0}^{T} \gamma^t \odot(a_{\leq t})(R_t - \gamma R_{t+1})$$

$$= \sum_{t=0}^{T} \gamma^t \odot(a_{\leq t})R_t - \sum_{t=0}^{T} \gamma^{t+1} \odot(a_{\leq t})R_{t+1}$$

Now we simply take a change of variables $t' = t + 1$ in the second term, relabeling the dummy variable immediately back to $t$:

$$J_{\odot} = \sum_{t=0}^{T} \gamma^t \odot(a_{\leq t})R_t - \sum_{t=1}^{T+1} \gamma^t \odot(a_{\leq t-1})R_t$$

$$= \sum_{t=0}^{T} \gamma^t \odot(a_{\leq t})R_t - \sum_{t=1}^{T+1} \gamma^t \odot(a_{< t})R_t$$

$$= \sum_{t=0}^{T} \gamma^t \odot(a_{\leq t})R_t - \sum_{t=0}^{T} \gamma^t \odot(a_{< t})R_t$$
$$+ \gamma^0 \odot(a_{<0})R_0 - \gamma^{T+1} \odot(a_{<T+1})R_{T+1}$$

$$= R_0 + \sum_{t=0}^{T} \gamma^t \left( \odot(a_{\leq t}) - \odot(a_{< t}) \right) R_t. \tag{6}$$

In the last line we have used that $\odot(a_{<0}) = \odot(\varnothing) = 1$, and that $R_{T+1} = 0$.

Now we have an objective formulated in terms of forwards-looking returns, that captures the dependencies on the sampling distribution through $\odot(a_{\leq t}) - \odot(a_{< t})$. Since this is just a re-expression of the DiCE objective (applied to a restricted class of SCGs with the requisite Markov property), we are still guaranteed that its derivatives will be unbiased estimators of the derivatives of our true objective, up to any order. The proof for the original DiCE objective is given by Foerster et al. [2018b]. Because $R_0$ carries no derivatives, we will omit it from the following estimators for clarity. Including it, however, ensures the convenient property that the objective still evaluates in expectation to the true return (as $\odot(a_{\leq t}) - \odot(a_{< t})$ always evaluates to zero).

We can now introduce value functions. $R_t$ is conditionally independent of each of $\odot(a_{\leq t})$ and $\odot(a_{< t})$ (as well as all their derivatives), conditioned on $a_{\leq t}, s_{\leq t}$. Because of the Markov property of our SCG, this is equivalent to conditional independence given $s_t, a_t$. If we consider the expectation of our new form of $J_{\odot}$, we can use this conditional independence to push the expectation over $a_{>t}, s_{>t}$ onto $R_t$. For a complete derivation please see the Supplementary Material. This is simply a critic value function, defined by $Q(s_t, a_t) = \mathbb{E}_{\pi}[R_t|s_t, a_t]$:

$$\mathbb{E}_{\pi}[J_{\odot}] = \mathbb{E}_{\pi}\left[ \sum_{t=0}^{T} \gamma^t \left( \odot(a_{\leq t}) - \odot(a_{< t}) \right) \mathbb{E}[R_t|s_t, a_t] \right]$$

$$= \mathbb{E}_{\pi}\left[ \sum_{t=0}^{T} \gamma^t \left( \odot(a_{\leq t}) - \odot(a_{< t}) \right) Q(s_t, a_t) \right] \tag{7}$$

Furthermore, a baseline that does not depend on $a_{\geq t}$ or $s_{>t}$ does not change the expectation of the estimator, as shown by the standard derivation reproduced in Schulman et al. [2015a]. In reinforcement learning, it is common to use the expected state value $V(s_t) = \mathbb{E}_{a_t}[Q(s_t, a_t)]$ as an approximation of the optimal baseline. The estimator may now use $A(s_t, a_t) = Q(s_t, a_t) - V(s_t)$ in place of $R_t$, further reducing its variance. We have now derived an estimator in terms of an advantage $A(s_t, a_t)$ that recovers unbiased estimates of derivatives of *any order*:

$$J_{\diamond} = \sum_t \gamma^t \left( \odot(a_{\leq t}) - \odot(a_{< t}) \right) A(s_t, a_t). \tag{8}$$

In practice, it is common to omit $\gamma^t$, thus optimising the undiscounted returns, but still using discounted advantages as a variance-reduction tool. See, e.g., the discussion in Thomas [2014].

## 3.1 Function approximation

In practice, an estimate of the advantage must be made from limited data. Inexact models of the critic value function (due to limited data, model class misspecification, or inefficient learning) introduce bias in the gradient estimators. As in the work of Schulman et al. [2015b], we may use combinations of sampled costs and estimated values to form advantage estimators that trade off bias and variance. However, thanks to our new estimator, which captures the full dependencies of the advantage on the sampling distribution, these trade-offs may be immediately applied to higher-order derivatives.

Approximate baseline value functions only affect the estimator variance. Careful choice of this baseline may nonetheless be of great significance (e.g., by exploiting the factorisation of the policy [Foerster et al., 2018c]). Our formulation of the objective extends such methods, as well as any future advances in advantage estimation at first order, to higher order derivatives.

## 3.2 Variance due to higher-order dependencies

Now that we have the correct form for the unbiased estimator, which uses proper variance-reduction strategies for computing the advantage, we may also trade off the bias and variance in estimates of higher-order derivatives that arises due to the full history of causal dependencies.

In particular, we propose to set a discount factor $\lambda \in [0, 1]$ on prior dependencies that limits the horizon of the past actions that are accounted for in the estimates of higher-order derivatives. Similarly to the way the MDP discount factor $\gamma$ reduces variance by constraining the horizon into the *future* that must be considered, $\lambda$ constrains how far into the *past* we consider causal dependencies that influence higher order derivatives.

First note that $\boxdot$ acting on any set of stochastic nodes $\mathcal{W}$ decomposes as a product: $\boxdot(\mathcal{W}) = \prod_{w \in \mathcal{W}} \boxdot(w)$. We now implement discounting by exponentially decaying past contributions:

$$J_\lambda = \sum_{t=0}^{T} \left( \prod_{t'=0}^{t} \boxdot(a_{t'})^{\lambda^{t-t'}} - \prod_{t'=0}^{t-1} \boxdot(a_{t'})^{\lambda^{t-t'}} \right) A_t. \tag{9}$$

This is our final objective, which we call "Loaded DiCE". The products over $\boxdot(\cdot)$ may be computed in the log-space of the action probabilities, transforming them into convenient and numerically stable sums. Algorithm 1 shows how the objective may easily be computed from an episode.

---

**Algorithm 1** Compute Loaded DiCE Objective

---

**Require:** trajectory of states $s_t$, actions $a_t$, $t = 0 \ldots T$
    $J \leftarrow 0$                                                 $\triangleright$ $J$ accumulates the final objective
    $w \leftarrow 0$                        $\triangleright$ $w$ accumulates the $\lambda$-weighted stochastic dependencies
    **for** $t \leftarrow 0$ to $T$ **do**
        $w \leftarrow \lambda w + \log(\pi(a_t|s_t))$                  $\triangleright$ $w$ has the dependencies including $a_t$
        $v \leftarrow w - \log(\pi(a_t|s_t))$                  $\triangleright$ $v$ has the dependencies excluding $a_t$
        `deps` $\leftarrow f(w) - f(v)$           $\triangleright$ $f$ applies the $\boxdot$ operator on the log-probabilities
        $J \leftarrow J + $ `deps` $\cdot A(s_t, a_t)$    $\triangleright$ The dependencies are weighted by the advantage $A(s_t, a_t)$
    **end for**
    **return** $J$

    **function** $f(x)$
        **return** $\exp(x - $ `stop_gradient`$(x))$
    **end function**

---

When $\lambda = 0$, this estimator resembles $J_{LVC}$, although it makes use of advantages. It may be low variance, but is biased regardless of the choice of advantage estimator. When $\lambda = 1$, we recover the estimator in (8), which is unbiased if the advantage estimator is itself unbiased. Intermediate values of $\lambda$ should be able to trade off bias and variance, as we demonstrate empirically in Section 4. Our new form of objective allows us to use $\lambda$ to reduce the impact of the high variance terms identified by Furmston et al. [2016] and Rothfuss et al. [2018] in a smooth way, rather than completely dropping those terms.

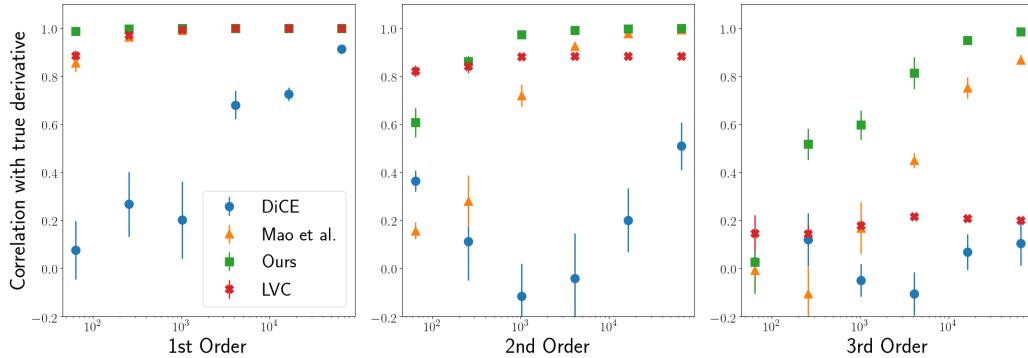

Figure 2: Convergence with increasing batch size of unbiased any-order estimators (DiCE, DiCE with the baseline of Mao et al. [2019], and Loaded DiCE). Also, LVC [Rothfuss et al., 2018], a low-variance but biased estimator.

## 4 Experiments

In this section we empirically demonstrate the correctness of our estimator in the absence of function approximation, and show how the bias and variance of the estimator may be traded off (a) by the choice of advantage estimator when only an approximate value function is available, and (b) by the use of our novel discount factor $\lambda$.

### 4.1 Bias and variance in any-order derivatives

To make the initial analysis simple and interpretable, we use small random MDPs with five states, four actions per state, and rewards that depend only on state. For these MDPs the discounted value may be calculated analytically, as follows.

$P^\pi$ is the state transition matrix induced by the MDP's transition function $P(s, a, s')$ and the tabular policy $\pi$, with elements given by

$$P_{ss'}^\pi = \sum_a P(s, a, s')\pi(a|s). \tag{10}$$

Let $P_0$ be the initial state distribution as a vector. Then, the probability distribution over states at time $t$ is a vector $p_{s_t} = (P^\pi)^t P_0$. The mean reward at time $t$ is $r_t = R^T p_{s_t}$, where $R$ is the vector of per-state rewards. Finally,

$$V^\pi = \sum_{t=0}^\infty \gamma^t r_t = R^T \sum_{t=0}^\infty (\gamma P_\pi)^t P_0$$
$$= R^T (I - \gamma P_\pi)^{-1} P_0. \tag{11}$$

This $V^\pi$ is differentiable wrt $\pi$ and may be easily computed with automatic differentiation packages. More details and code can be found in the Supplementary Material.

**A low-variance, unbiased, any-order estimator.** Figure 2 shows how the correlation between estimated and true derivatives changes as a function of batch size, for up to third order. We compare the original DiCE estimator to Loaded DiCE, and the objective proposed by Mao et al. [2019] which incorporates only a baseline. For Loaded DiCE, we use $A^{GAE}$ with $\tau = 0$, the exact value function, and $\lambda = 1$ (so as to remain unbiased). As these are all unbiased estimators, they will converge to the true derivatives with a sufficiently large batch size. However, when using an advantage estimator with the exact value function, the variance may be dramatically reduced and the estimates converge much more rapidly. We also show the performance of LVC [Rothfuss et al., 2018]. At first order it matches exactly the estimator of Mao et al. [2019], but underperforms Loaded DiCE because it does not use the advantage. At higher orders, it is low variance but biased, as expected.

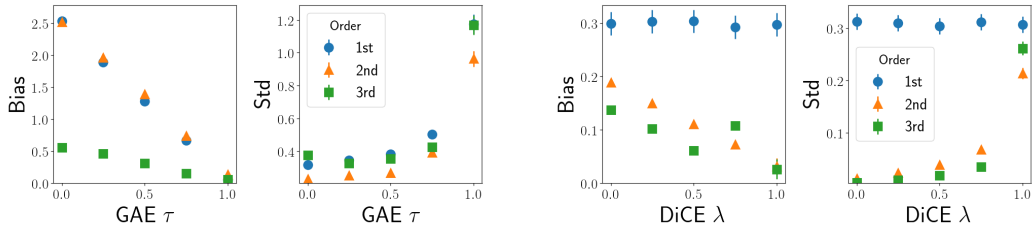

(a) Low $\tau$ produces low variance estimates at the cost of high bias. The effect holds at all orders of derivatives.

(b) High $\lambda$ considers the full past to produce low-bias high-variance estimators, low $\lambda$ discounts the past. First order gradients are unaffected.

Figure 3: Trading off bias and variance with $\tau$ and $\lambda$ in a small MDP.

**Trading off bias and variance with advantage estimation.**  Figure 3a shows the bias and standard deviation of estimated derivatives using a range of $\tau$, and an inexact value function (we perturb the true value function with Gaussian noise for each state to emulate function approximation). The effect of the choice of advantage estimator trades off bias and variance not only at the first order, but in any-order derivatives.

**Trading off bias and variance by discounting causes.**  Figure 3b shows the bias and standard deviation of estimated derivatives using a range of $\lambda$. To isolate the effect of $\lambda$ we use the exact value function and $\tau = 0$, so the absolute bias and variance are lower than in figure 3a. First-order derivatives are unaffected by $\lambda$, as expected. However, in higher-order derivatives, $\lambda$ strongly affects the bias and variance of the resulting estimator. There is an outlier at $\lambda = 0.75$ for third order derivatives – there is no guarantee of monotonicity in the bias or variance, but we found such outliers rarer at second than third order, and appearing as artefacts of particular MDPs.

## 4.2  Meta reinforcement learning with MAML and Loaded DiCE

We now apply our new family of estimators to a pair of more challenging meta-reinforcement-learning problems in continuous control, following the work of Finn et al. [2017]. The aim of their Model-Agnostic Meta-Learning (MAML) is to learn a good initialisation of a neural network policy, such that with a single (or small number of) policy gradient updates on a batch of training episodes, the policy achieves good performance on a task sampled from some distribution. Then, in meta-testing, the policy should be able to adapt to a new task from the same distribution. MAML is theoretically sound, but the original implementation neglected the higher order dependencies induced by the RL setting [Rothfuss et al., 2018, Stadie et al., 2018].

The approach is to sample a number of tasks and adapt the policy in an inner-loop policy-gradient optimisation step. Then, in the outer loop, the initial parameters are updated so as to maximise the returns of the post-adaptation policies. The outer loop optimisation depends on the post-adaptation parameters, which depend on the gradients estimated in the inner loop. As a result, there are important higher-order terms in the outer loop optimisation. Using the correct estimator for the inner loop optimisation can therefore impact the efficiency of the overall meta-training procedure as well as the quality of the final solution.

For the inner-loop optimisation, we use our novel objective, with a range of values for $\tau$ and $\lambda$. We sweep a range of $\tau$ with fixed $\lambda = 0$, and then sweep a range of $\lambda$ using the best value found for $\tau$. For the outer-loop optimisation, we use a vanilla policy gradient with a baseline. The outer-loop could use any other gradient-based policy optimisation algorithm, but we choose a simple version to isolate, to some extent, the impact of the inner loop estimator.

Figure 4 shows our results. In the CheetahDir task, if $\tau$ is too high the estimator is too high variance and performance is bad. $\tau$ is less impactful in the CheetahVel task. We note that in these tasks, episodes are short, $\gamma$ is low, and the value functions are simple linear functions fit to each batch of data as in Finn et al. [2017]. These factors which would all favor a high $\tau$. With higher variance returns or better value functions, relying more heavily on the learned value function (by using a lower $\tau$) may be effective.

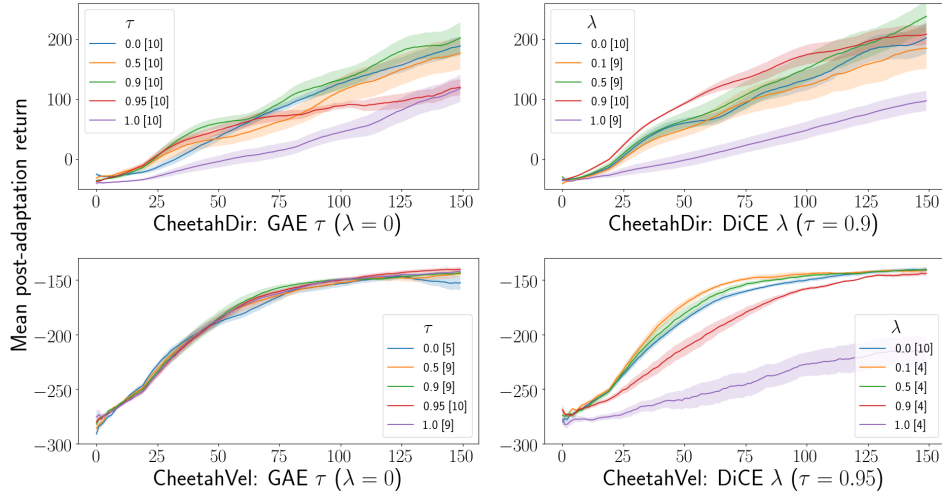

Figure 4: Trading off bias and variance with $\tau$ and $\lambda$ in meta-reinforcement-learning. We report the mean and standard error (over [#runs]) of the post-adaptation returns, smoothed with a moving average over 10 outer-loop optimisations.

In both environments, $\lambda = 1.0$ leads to too high variance. The unbiased ($\lambda = 1.0$) version of our objective may also be more valuable when value functions are better and can be used more effectively to mitigate variance. In CheetahVel, noticeably faster learning is achieved with a low but non-zero $\lambda$. The analysis of Furmston et al. [2016] indicates that the magnitude of the higher-order terms discounted by $\lambda$ will in many cases become small as the policy approaches a local optimum. This is consistent with our empirical finding here that non-zero $\lambda$ may learn faster but plateaus at a similar level. Figure 5 in the appendix shows results on the AntVel task, in which $\lambda$ is a more important factor than $\tau$. We conclude that Loaded DiCE provides meaningful control of the higher-order estimator with significant impact on a realistic use-case.

## 5  Conclusion

In this work, we derived a theoretically sound objective which can apply general advantage functions to the estimation of any-order derivatives in reinforcement-learning type sequential problems. In the context of function approximation, this objective unlocks the ability to trade off bias and variance in higher order derivatives. Importantly, like the underlying DiCE objective, our single objective generates estimators for any-order derivatives under repeated automatic differentiation. Further, we propose a simple method for discounting the impact of more distant causal dependencies on the estimation of higher order derivatives, which allows another axis for the trade-off of bias and variance. Empirically, we use small random MDPs to demonstrate the behaviour of the bias and variance of higher-order derivative estimates, and further show its utility in meta-reinforcement-learning.

We are excited for other applications in meta-learning, multi-agent learning and higher-order optimisation which may be made possible using our new objective. In future work, we also wish to revisit our choice of $\lambda$-discounting, which is a heuristic method to limit the impact of high-variance terms. Further theoretical analysis may also help to identify contexts in which higher-order dependencies are important for optimisation. Finally, it may even be possible to meta-learn the hyperparameters $\tau$ and $\lambda$ themselves.

### Acknowledgments

We thank Maruan Al-Shedivat and Minqi Jiang for valuable discussions. This work was supported by the UK EPSRC CDT in Autonomous Intelligent Machines and Systems. This project has received funding from the European Research Council (ERC) under the European Union's Horizon 2020 research and innovation programme (grant agreement number 637713).

## Footnotes

*Correspondence to `gregory.farquhar@cs.ox.ac.uk`

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

# 6 Derivation of value function formulation

We start out with the $J_\diamond$ objective:

$$J_\diamond = \sum_{t=0}^{T} \gamma^t \left( \boxdot(a_{\leq t}) - \boxdot(a_{<t}) \right) R_t. \tag{12}$$

We evaluate this objective by taking an expectation over the trajectories $\tau$ as induced by the policy $\pi$. Here $\tau$ is a complete sequence of states, actions and rewards, $\tau = \{s_0, a_0, r_1, .., s_T, a_T\}$. For convenience in the following derivation we have defined the reward, $r_{t+1}$, to be indexed by the next time step, after action $a_t$ was taken. This ensures that partial trajectories (e.g. $\tau_{>t}$) correctly keep rewards after the actions that cause them. Note that $R_t = \sum_{k=0}^{T-t} \gamma^k r_{t+k+1}$ depends only on $\tau_{>t}$. The expectation of our objective is given by:

$$\mathbb{E}_\pi[J_\diamond] = \sum_\tau P(\tau) J_\diamond(\tau) \tag{13}$$

$$= \sum_\tau P(\tau) \left( \sum_{t=0}^{T} \gamma^t \left( \boxdot(a_{\leq t}) - \boxdot(a_{<t}) \right) R_t \right) \tag{14}$$

$$= \sum_{t=0}^{T} \gamma^t \left( \sum_\tau P(\tau) \left( \boxdot(a_{\leq t}) - \boxdot(a_{<t}) \right) R_t \right) \tag{15}$$

$$= \sum_{t=0}^{T} \gamma^t J_t \tag{16}$$

We note that for each time step the term, $J_t$ is of the form:

$$J_t = \sum_\tau P(\tau) f(\tau_{\leq t}) g(\tau_{>t}), \tag{17}$$

where $f(\tau_{\leq t}) = \left( \boxdot(a_{\leq t}) - \boxdot(a_{<t}) \right)$ and $g(\tau_{>t}) = R_t$.

Next we use:

$$P(\tau) = P(\tau_{\leq t}) P(\tau_{>t} | \tau_{\leq t}) \tag{18}$$

$$= P(\tau_{\leq t}) P(\tau_{>t} | s_t, a_t), \tag{19}$$

where in the last step we have used the Markov property. Substituting we obtain:

$$J_t = \sum_\tau P(\tau_{\leq t}) P(\tau_{>t} | s_t, a_t) f(\tau_{\leq t}) g(\tau_{>t}) \tag{20}$$

$$= \sum_{\tau_{\leq t}} P(\tau_{\leq t}) f(\tau_{\leq t}) \sum_{\tau_{>t}} P(\tau_{\geq t} | s_t, a_t) g(\tau_{>t}) \tag{21}$$

If we substitute back for $g$ and $f$ we obtain:

$$J_t = \sum_{\tau_{\leq t}} P(\tau_{\leq t}) \left( \boxdot(a_{\leq t}) - \boxdot(a_{<t}) \right) \sum_{\tau_{>t}} P(\tau_{\geq t} | s_t, a_t) R_t \tag{22}$$

$$= \sum_{\tau_{\leq t}} P(\tau_{\leq t}) \left( \boxdot(a_{\leq t}) - \boxdot(a_{<t}) \right) \mathbb{E}[R_t | s_t, a_t] \tag{23}$$

$$= \sum_{\tau_{\leq t}} P(\tau_{\leq t}) \left( \boxdot(a_{\leq t}) - \boxdot(a_{<t}) \right) Q(s_t, a_t) \tag{24}$$

$$\tag{25}$$

Putting all together we obtain the final form:

$$\mathbb{E}_\pi[J_\diamond] = \mathbb{E}_\pi \left[ \sum_{t=0}^{T} \gamma^t \left( \boxdot(a_{\leq t}) - \boxdot(a_{<t}) \right) Q(s_t, a_t) \right] \tag{26}$$

# 7 Experimental details

## 7.1 Random MDPs

We use the `mdptoolbox.example.rand()` function from PyMDPToolbox to generate random MDP transition functions with five states and four actions per state.

The reward is a function only of state, and is sampled from $\mathcal{N}(5, 10)$. We use $\gamma = 0.95$. When sampling for the stochastic estimators, we use batches of 512 rollouts of length 50 steps unless the batch size is otherwise specified.

We only compute higher order derivatives of the derivative of the first parameter at each order, to save computation.

For the sweeps over $\lambda$ and $\tau$ we use 200 batches for each value of $\lambda$ or $\tau$. To simulate function approximation error in our analysis of the impact of $\tau$, we add a gaussian noise with standard deviation 10 to the true value function.

## 7.2 MAML experiments

We use the following hyperparameters for our MAML experiments:

| Parameter | Value |
|---|---|
| $\gamma$ | 0.97 |
| hidden layer size | 100 |
| number of layers | 2 |
| task batch size | 20 trajectories |
| meta batch size | 40 tasks |
| inner loop learning rate | 0.1 |
| outer loop optimiser | Adam |
| outer loop learning rate | 0.0005 |
| outer loop $\tau$ | 1.0 |
| reward noise | Uniform(-0.01, 0.01) at each timestep |

We also normalise all advantages in each batch (per task).

Figure 5 shows some additional experiments on the AntVel MuJoCo task. In this domain, $\lambda$ is a more important factor than $\tau$.

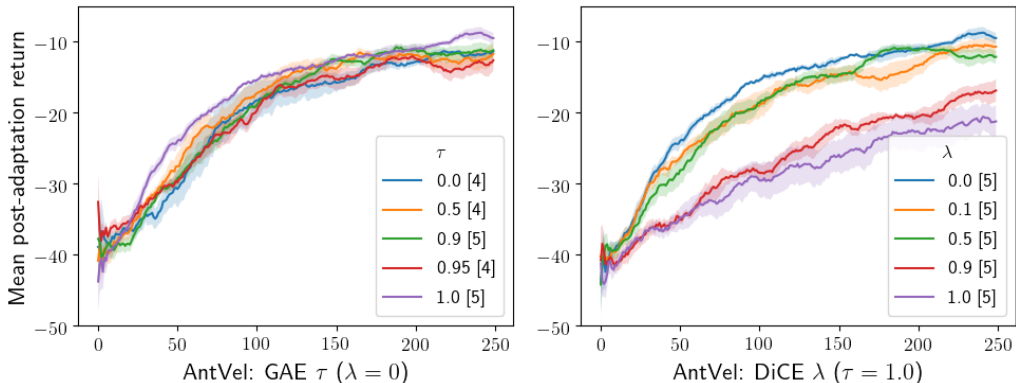

Figure 5: Trading off bias and variance with $\tau$ and $\lambda$ in the ant-velocity task. We report the mean and standard error (over [#runs]) of the post-adaptation returns, smoothed with a moving average over 10 outer-loop optimisations

