[Reviews · NeurIPS 2019]

Reviewer 1



The DiCE gradient estimator [1] allows the computation of higher-order derivatives in stochastic computation graphs. This may be useful in contexts such multi-agent learning or meta-RL where the proper application of methods such as MAML require the computation of second-order derivatives. The current paper extends DiCE and derives a more general objective that allows integration of the advantage A(s_t, a_t) = Q(s_t, a_t) - V(s_t) in order to control for the variance while providing unbiased estimates. The advantage can be approximated by trading off variance for bias using parametric function approximators and methods such as Generalized Advantage Estimation (GAE). Moreover, the authors propose to further control the variance of the higher-order gradients by discounting the impact past actions on the current advantage, thus limiting the range of causal dependencies. This paper is well executed: it is well written, technically sound and potentially impactful. The method is tested on a toy domain, which highlights that the new estimator is more efficient than baselines at estimating the true higher-order derivatives, and in a meta-RL task. I'd suggest the authors to strengthen and detail a bit more their experimental setting: - In Section 4.1, I'd suggest the authors to add a few lines on how the ground-truth higher order gradients are computed in the small MDP. I couldn't find that information in the supplementary material. - In Figure 2, the convergence plots wrt to the batch size are nice. Are you using a \lambda = 1 ? - In Section 4.2, the authors test on Half-Cheetah. Citing footnote 2 in [3] "half-cheetah is an example of a weak benchmark–it can be learned in just a few gradient step from a random prior. It can also e solved with a linear policy. Stop using half-cheetah." I wonder if the authors can find some other more challenging tasks that can show the empirical superiority of their method. - In Figure 4, I think you are using Loaded-DiCE with the best \tau found for GAE. If that's the case, you should write it down. - In Section 4.2, it'd be better to add a few more baselines ? Maybe [2] ? Pros: + Address an important problem + Well written and technically sound. + Empirically validated on a toy setting. Cons: - The range of realistic use-case applications is rather limited. [1] https://arxiv.org/abs/1802.05098 [2] https://arxiv.org/pdf/1803.01118.pdf

Reviewer 2



This paper extends the DiCE objective in a way that allows automatic differentiation to produce low-variance unbiased estimators of derivatives of any order. This is done in a way that can incorporate any advantage estimator, and shown to be accurate and valuble in practice through two experiments. As someone who knows little about this field, I found the paper well-written, with sufficient references to and background of existing literature. What's known and what's novel is clearly pointed out, and the motivation for Loaded DiCE is well justified. The extension from DiCE appears efficient to compute, even though much more powerful. Figures 3 and 4 show substantial improvement over the previous state of the art. The work seems to be of high quality and is presented clearly enough to be read by non-RL researchers. Given my limited knowledge of the area, I'll defer a careful assessment of the significance and originality of the work to other reviewers.

Reviewer 3



The paper sets out to derive an arbitrary order score function estimator that can balance variance and bias. With this in mind, I think that the strenghts and weaknesses of the paper are: + well written, clear presentation + clear derivation of new algorithm - the empirical analysis is somewhat weak Other minor comments - the subfigures in Figure 2 are nearly unreadable, expecially in b&w - the authors might want to consider "conjugate MDPs" and how they relate to their work UPDATE: While I still do not feel perfectly confident in my assessment, I find the author's rebuttal to be satisfying. I increase my score.

[Author Response · NeurIPS 2019]

We would like to thank the reviewers for their helpful feedback, and for their positive comments regarding the originality and significance of our new objective, as well as the clarity of its derivation and exposition. Please find our responses to specific reviewer questions or comments below.

**Analytical derivatives of MDP value (@R1)**

The probability distribution over states at time $t$ is $p(s_t) = (P_\pi)^t P_0$, where $P_\pi$ is the state transition matrix defined by the MDP's transition function and the (tabular) policy $\pi$, and $P_0$ is the initial state distribution. The mean reward at time $t$ is $r_t = R^T p(s_t)$, where $R$ is the vector of per-state rewards. Then $V^\pi = \sum_{t=0}^\infty \gamma^t r_t = R^T \sum_{t=0}^\infty (\gamma P_\pi)^t P_0 = R^T (I - \gamma P_\pi)^{-1} P_0$. This $V^\pi$ is differentiable wrt $\pi$ and may be easily computed with automatic differentiation packages. We will clarify this formulation in the paper.

**Relationship to Conjugate MDPs (@R3)**

Conjugate MDPs are an interesting framework for learning useful abstractions. In their original formulation (and in the most obvious extensions to deep actor-critic methods), only first-order derivatives would be required. However, we could imagine a similar framework in which the co-agent is optimised with some awareness of the learning process of the primary agent (e.g. is optimised such that the primary agent performs well after some additional gradient-based learning), which could make use of higher-order derivatives and therefore our new objective. We regard this as an exciting avenue for future work, and can certainly discuss further potential applications in the paper.

**Figures**

**Legibility (@R3).** Thank you for your feedback – we will improve the legibility of all figures with attention to B&W.
**Figure 2 (@R1).** "Ours" in Figure 2 is indeed using $\lambda = 1$, so as to compare against the other unbiased estimators. We will clarify this in the text.
**Figure 4 (@R1).** We are indeed using the best $\tau$ found for GAE when running with $\lambda = 0$. We will clarify this also.

**MAML experiments**

We would like to emphasise that the main contribution of our work is the derivation of an objective that produces a family of useful estimators, and that we look forward to future work to explore the full range of possible applications for higher order derivatives (which extend well beyond MAML-style meta-RL). We know of one research group already making use of Loaded DiCE for their work on multi-agent learning, and think it would be a valuable tool to share with the whole NeurIPS community. To respond directly to R3, we do not feel that wanting more (unspecified) experiments justifies a score of 4, when the review does not contain a single other substantive criticism.

That being said, we do want to evidence the practical utility of our objective, and will address some of the specific comments on the final part of our empirical study here.

**Half Cheetah (@R1).** Half Cheetah is indeed a somewhat limited benchmark. We chose it because it trains quickly, and the base algorithm required no additional tuning to work out of the box (unlike other environments, where we found existing implementations would perform unreliably or require very large amounts of training).

We ran some additional experiments on the Ant MuJoCo domain, with results shown here. In this domain, $\lambda$ is a more important factor than $\tau$. We also note that, following existing implementations, our value function is a simple linear function of some handcrafted features. We expect that the utility of our objective will be more pronounced as researchers use stronger (and themselves meta-learned) value functions and tackle harder domains, but this will require further research in other aspects of meta-learning for RL. We also believe that such research will be facilitated by the use of our objective!

**Baselines (@R1).** The proposed baseline method (E-MAML) is mathematically equivalent to DiCE. DiCE is simply an objective that makes the calculation of the E-MAML estimator much simpler, and generalises to other applications of higher-order derivatives (rather than just MAML-RL). As such, our proposed method, which generalises DiCE, will have equal or better performance. Since our method also generalises LVC, we believe the range of comparisons is fairly comprehensive with respect to the choice of higher-order estimator. We will include for completeness experiments without any baseline for variance reduction (which substantially underperform in all cases – e.g. no more than -70 achieved on the AntVel task inset above).

It would certainly be interesting to explore the utility of our objective by combining it with a more elaborate setup involving more complex value functions and a more advanced base learning algorithm (like PPO). However, we wanted to keep the setup simple to focus on our key point: trading off bias and variance in estimators of higher-order derivatives is important, and our new objective gives a principled and straightforward way to do so.

[Meta-Review · NeurIPS 2019]

This paper presents novel methodology in combination with automatic differentiation, that yields unbiased and low-variance estimators of derivatives at any order. It appears potentially to be widely useful, and the exposition is clear to understand. The reviewers and I seem to be in general agreement in liking the paper. Reviewer 1 wrote a thorough review touching on many aspects of the paper. The overall score was 7, and his bottom line positives were: "This paper is well executed: it is well written, technically sound and potentially impactful." The main bottom line negative was "The range of realistic use-case applications is rather limited." Reviewer 2 also gave overall score of 7, and agreed with much of R1's comments. For example: "I found the paper well-written, with sufficient references to and background of existing literature. What's known and what's novel is clearly pointed out, and the motivation for Loaded DiCE is well justified ... The work seems to be of high quality and is presented clearly enough to be read by non-RL researchers." However, due to R2's limited knowledge of this topic, he deferred a careful assessment of the significance and originality of the work to other reviewers. As a result, the confidence was only 2. Reviewer 3 originally gave an overall score of 5. However, after reading the reviews and author feedback, he increased his score to 6. R3 also gave a low-confidence score of 1 due to lack of expertise in this topic. My take is that the paper gives a nice contribution, is correct and easily accessible, and may prove to be widely useful, and providing opportunities to build further on this line of research. With overall scores of (7, 7, 6) the paper is in a range where it should have a decent chance of being accepted at NeurIPS. However, due to the very low confidence scores of R2 and R3, the Senior Area Chair and I decided to seek out an additional last-minute reviewer who is quite familiar with previous work that provides the basis of the current submission. Due to lack of time, this reviewer provided the following brief comments: "I cannot say for the correctness of the paper, but I think that there is a useful contribution in the paper: that of using the Dice trick with control variables (GAE). Afaik, the dice paper only considers plain finite-horizon likelihood ratio estimator.  The experiments show a plot of the standard deviation as a function of the order, and a maml-experiment with confidence bands around their curves. The experimental methodology looks good to me." With this additional input, the Senior Area Chair and I agree that there is now sufficient confidence to agree with the unanimous recommendations of Accept from the initial three reviewers.